# SpatioTemporal-GRPO: Post-Training Large Multimodal Models for Video QA

## Abstract

We introduce SpatioTemporal-GRPO (ST-GRPO), a novel extension of the GRPO algorithm for video question answering. ST-GRPO addresses a limitation of standard GRPO: when all responses in a group have similar correctness, the low reward variance gives the model an uninformative signal for improvement. Our method overcomes this by generating multiple spatiotemporal variants of a video to serve as complementary inputs. Unlike standard GRPO, which only groups textual responses, ST-GRPO forms groups across both textual and spatiotemporal variants. This increases reward variance within each group, providing a more informative signal for learning. To ensure these visual variations are meaningful, we propose an importance-based grouping strategy. This approach computes per-frame relevance scores using cross-modal embeddings, prioritizing frames that carry higher semantic weight relative to the question. This question-aware method ensures our spatiotemporal groups are informed by the relevant visual cues for each query. Our experiments demonstrate consistent improvements across six challenging video understanding benchmarks, including VideoMME, TempCompass, VideoMMMU, MMVU, VSI-Bench, and PerceptionTest, showing that incorporating structured visual diversity into reinforcement learning provides a more effective approach for learning from spatiotemporal cues in video question answering.

## 1 Introduction

Recent advances in large multimodal models (LMMs) have driven rapid progress in video question answering (VQA) (Li et al., 2024; Bai et al., 2025; Cheng et al., 2024), where models must jointly reason over visual and textual inputs to answer complex queries about videos. While supervised fine-tuning has been the dominant strategy for improving performance, its reliance on large-scale annotated datasets limits scalability. Post-training reinforcement learning (RL) has emerged as a promising alternative, enabling models to refine outputs without costly human supervision. In this context, Group Relative Policy Optimization (GRPO) (Shao et al., 2024; Li et al., 2025a) has emerged as a popular post-training method for aligning model outputs with task-specific reward signals, such as human preference feedback or correctness in question answering. By sampling multiple responses per input and normalizing rewards within each group, GRPO encourages models to favor better responses.

RL-based post-training has already demonstrated strong gains in language domains such as mathematics, where it improves both step-by-step logical consistency and final-answer accuracy (Shao et al., 2024). More recently, these methods have begun to extend into vision-language domains, where multimodal reasoning requires not only textual alignment but also robust modeling of spatial and temporal cues (Li et al., 2025a; Xue et al., 2025). However, despite GRPO's efficiency compared to other RL-based approaches, current formulations remain restricted to text-only settings, leaving the rich spatiotemporal structure of video inputs underutilized.

In this work, we introduce **SpatioTemporal-GRPO (ST-GRPO)**, a variant of GRPO tailored for video question answering (VQA). ST-GRPO extends GRPO into the visual domain by generating multiple spatiotemporal variants of the same video, which are used as complementary inputs alongside the text-based ones in GRPO. As illustrated in Fig. 1, unlike standard GRPO which only groups responses based on textual variations, ST-GRPO forms groups across both textual and spatiotempo-

Figure 1: Comparison between GRPO and the proposed ST-GRPO for video question answering (VQA). (a) In standard GRPO, the model generates multiple responses for a given video–question pair and updates the policy using group-wise rewards. (b) ST-GRPO introduces SpatioTemporal (ST) transformations to create diverse variants of the same video. Each variant forms an output group (e.g., Group #1 shown in blue), enabling dual grouping strategies that leverage both temporal diversity and response diversity.

ral variants, thereby explicitly incorporating visual diversity into the optimization process. This design directly addresses a known bias in GRPO: when the variance of rewards within a group is close to zero—for instance, when most responses are uniformly correct or incorrect—the learning signal becomes unstable and may lead to disproportionate updates. While Dr.GRPO (Liu et al., 2025) alleviates this issue by removing variance normalization, it under-weights rare correct responses within predominantly incorrect groups.

To further ensure that visual variation is semantically meaningful, we introduce an **importance-based grouping** strategy. Instead of treating all frames equally, we compute per-frame importance scores based on their relevance to the question. Using a multimodal model, we extract cross-modal embeddings to identify frames that carry higher semantic weight. By prioritizing these frames, ST-GRPO forms spatiotemporal groups that are directly informed by the question, ensuring the model learns from the most semantically relevant visual cues for answering. Together, spatiotemporal grouping and importance-based grouping enable stronger performance in VQA by providing a more effective way to learn from relevant spatiotemporal cues. Our experiments show consistent improvements across challenging benchmarks, including VideoMME (Fu et al., 2024), TempCompass (Liu et al., 2024), VideoMMMU (Hu et al., 2025), MMVU (Zhao et al., 2025), VSI-Bench (Yang et al., 2024), and PerceptionTest (Patraucean et al., 2023).

Our contributions are as follows:

- We propose **SpatioTemporal-GRPO (ST-GRPO)**, the first extension of GRPO to video-language domains, which leverages spatiotemporal variants of video inputs to form visual groups that enhance reward variance.

- We introduce an **importance-based grouping** mechanism that employs question-guided frame selection, ensuring that visual diversity is semantically meaningful and aligned with the query.

- We analyze the bias of GRPO under low reward variance and demonstrate that ST-GRPO mitigates both the exaggerated updates in GRPO and the underweighting bias in Dr.GRPO, leading to consistent improvements across six challenging VQA benchmarks.

## 2 RELATED WORK

### 2.1 LARGE MULTIMODAL MODELS (LMMs)

Large multimodal models (LMMs) have rapidly advanced in recent years, extending large language models with visual encoders to handle images and videos in complex reasoning tasks. Early progress centered on image-based multimodal modeling, such as visual instruction tuning (Liu et al., 2023) and the consolidated LLaVA-OneVision series (Li et al., 2024). These approaches improved OCR, grounding, and reasoning, leading to stronger baselines on vision-language tasks. More recently, research has shifted toward video understanding. Models such as Video-LLaMA (Zhang et al., 2023),

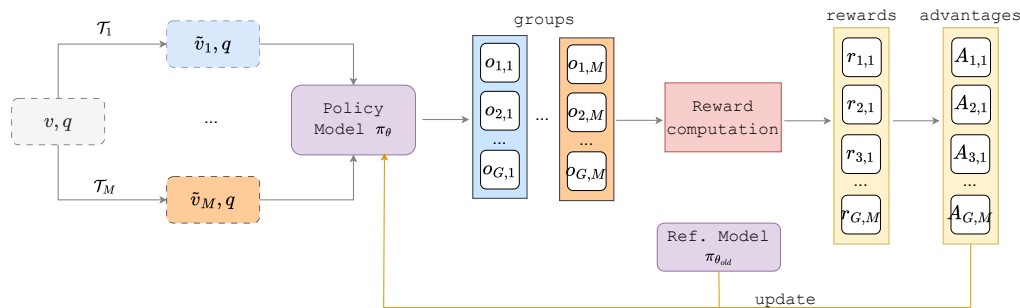

Figure 2: **Overview of our ST-GRPO**. Given a video input $v$ and question $q$, a set of transformations $\{\mathcal{T}_i\}_{i=1}^{M}$ is applied to generate video variants $\{\tilde{v}_i\}_{i=1}^{M}$. Each variant is paired with the question to form input tuples $(\tilde{v}_i, q)$. The policy model produces $G \times M$ responses $\{o_{i,j}\}_{i=1,j=1}^{G,M}$, each evaluated by a reward model to obtain $\{r_{i,j}\}_{i=1,j=1}^{G,M}$. These rewards are grouped both across video variants and within sampling groups.

Video-LLaVA (Lin et al., 2023), and Video-ChatGPT (Maaz et al., 2023) extend instruction-tuned LMMs to temporal reasoning. Benchmarks like Video-MME (Fu et al., 2024), Video-MMMU (Hu et al., 2025), VSI-Bench (Yang et al., 2024), and TempCompass (Liu et al., 2024), as well as Perception Test (Patraucean et al., 2023), highlight the challenges of fine-grained spatiotemporal reasoning. Despite strong progress, open models often trail proprietary systems such as GPT-4o (Hurst et al., 2024) and Gemini 1.5 (Gemini Team, 2024), whose detailed training methods remain undisclosed.

## 2.2 POST-TRAINING FOR LMMS

Reinforcement learning (RL) is a widely used post-training paradigm for aligning large language models with human preferences (Stiennon et al., 2020; Ouyang et al., 2022; Rafailov et al., 2023). Methods such as PPO (Schulman et al., 2017), DPO Rafailov et al. (2023), and GRPO (Shao et al., 2024) have significantly improved reasoning capabilities in LLMs. Building on this progress, RL has been increasingly applied to LMMs. In particular, GRPO has been adopted for multimodal alignment as it enables group-wise optimization over multiple responses.

Recent works adapt GRPO to temporal reasoning in videos. ArrowRL (Xue et al., 2025) introduces a "reverse reward" mechanism that encourages models to distinguish forward versus reversed video sequences, instilling Arrow-of-Time awareness. Video-R1 (Feng et al., 2025) proposes T-GRPO, contrasting reasoning performance on ordered versus shuffled frames to enhance temporal perception. DeepVideo-R1 (Li et al., 2025b) further introduces Reg-GRPO, a regression-style variant of GRPO, together with difficulty-aware video augmentations to mitigate vanishing advantages. Complementary efforts explore reward shaping and sharing strategies: R1-ShareVL (Yao et al., 2025) expands each query into variants and hierarchically shares advantages, while VideoChat-R1 (Li et al., 2025c) conducts multi-task reinforcement fine-tuning for spatiotemporal perception.

While these methods advance multimodal alignment, they largely retain the standard GRPO formulation and address temporal perception primarily through reward engineering or auxiliary augmentations. ArrowRL and Video-R1 rely on handcrafted contrastive rewards based on forward/reverse or ordered/shuffled variants, while DeepVideo-R1 uses difficulty-aware augmentations outside of the grouping mechanism. In contrast, our ST-GRPO extends GRPO by integrating *spatiotemporal video variants* directly into the grouping strategy. By forming groups that combine these variants with a question-aware importance-based selection of frames, ST-GRPO enriches reward variance and provides more reliable learning signals in low-variance regimes. This structured grouping strategy leads to consistent gains over prior GRPO-based post-training approaches on video question answering benchmarks.

## 3 METHOD

DeepSeekMath (Shao et al., 2024) introduced Group Relative Policy Optimization (GRPO), a variant of Proximal Policy Optimization (PPO) (Schulman et al., 2017). Unlike PPO, GRPO eliminates the need for a separate value function by estimating the advantage directly from rewards of multiple sampled outputs for the same input.

Specifically, in the case of video question answering (VQA), given a question $q$ and a video input $v$, the model samples multiple responses $\{o_i\}_{i=1}^G$ from a large multimodal model. Each output receives a reward $\{r_i\}_{i=1}^G$, and the policy is then optimized using the GRPO objective:

$$
J_{\text{GRPO}}(\theta) = \frac{1}{G} \sum_{i=1}^G \left[ \min\left( \frac{\pi_\theta(o_i \mid q, v)}{\pi_{\theta_{\text{old}}}(o_i \mid q, v)} A_i, \ \text{clip}\left( \frac{\pi_\theta(o_i \mid q, v)}{\pi_{\theta_{\text{old}}}(o_i \mid q, v)}, \ 1 - \epsilon, \ 1 + \epsilon \right) A_i \right) \right] \\ - \beta \, \mathbb{D}_{\text{KL}}\big[\pi_\theta \,||\, \pi_{\text{ref}}\big]
$$

(1)

where $A_i$ denotes the normalized advantage of output $o_i$, defined as

$$
A_i = \frac{r_i - \text{mean}(r_1, \ldots, r_G)}{\text{std}(r_1, \ldots, r_G)}.
$$

(2)

and $\mathbb{D}_{\text{KL}}$ is the KL divergence between the trained policy $\pi_\theta$ and the reference policy $\pi_{\text{ref}}$.

**Low reward variation in GRPO**  The group-relative policy optimization (GRPO) advantage, as defined in Eq. 2, is obtained by normalizing each reward within a group using its mean and standard deviation. This formulation compares multiple responses to the same input and encourages the model to favor better responses, i.e., the correct answer in a question–answering setting. However, GRPO exhibits a bias when the variance of rewards is close to zero—such as when most responses in a group are either all correct or all incorrect. In this case, the standard deviation becomes very small, inflating the advantages and causing overly strong updates despite a limited discriminative signal. To mitigate this, Dr.GRPO (Liu et al., 2025) removes the standard deviation term, yielding more stable estimates in low-variance cases. Yet this introduces another bias: when a few correct responses appear in an otherwise incorrect group, their contribution is down-weighted, weakening a valuable learning signal. Increasing the reward variance alleviates both issues: in GRPO, it prevents exaggerated updates, while in Dr.GRPO, it makes rare correct responses stand out more clearly. In the context of video question answering, where the input consists of both a video and a textual query, the video can be leveraged to construct visual groups as complementary to the original groups in GRPO. By providing the model with different variants of the same video input, these visual groups increase reward variance, thereby mitigating the complementary biases of both GRPO and Dr.GRPO.

## 4 SPATIOTEMPORAL-GRPO

Large Multimodal Models (LMMs) process both video and question inputs to generate answers grounded in visual content. Post-training methods such as Group Relative Policy Optimization (GRPO) (Xue et al., 2025; Fan et al., 2025) have been widely adopted to better align model outputs with task-specific objectives. GRPO leverages the generative capacity of LMMs by producing multiple responses per input and computing group-wise advantages to guide optimization. In the context of video-based tasks, the visual modality offers an additional source of structure that can be exploited for grouping. To this end, we propose **SpatioTemporal-GRPO (ST-GRPO)**, which extends GRPO by introducing spatiotemporal groups (ST-Group). As shown in Fig. 2, ST-GRPO implements dual grouping strategies—across video variants and within sampling groups—by generating diverse variants of the same video input. This complementary grouping enriches the reward variance and improves the effectiveness of policy optimization.

Given a video input $v$ and question $q$, we apply a set of $M$ transformations $\{\mathcal{T}_i\}_{i=1}^M$ to generate a set of video variants $\{\tilde{v}_i\}_{i=1}^M$ where $\tilde{v}_i = \mathcal{T}_i(v)$. Each variant is paired with the question to form input

tuples $\{(\tilde{v}_i, q)\}_{i=1}^M$. The model then generates $G \times M$ responses $\{o_{i,j}\}_{i=1,j=1}^{G,M}$ with corresponding rewards $\{r_{i,j}\}_{i=1,j=1}^{G,M}$. This approach enables dual grouping strategies—across different video variants and within sampling groups—thereby enriching the diversity of the reward distribution for more effective reinforcement learning. The objective of ST-GRPO is as follows where we omit the KL divergence term for simplicity:

$$J_{\text{ST-GRPO}}(\theta) = \frac{1}{G} \frac{1}{M} \sum_{i=1}^{G} \sum_{j=1}^{M} \left[ \min\left( \frac{\pi_\theta(o_{i,j} \mid q, v)}{\pi_{\theta_{\text{old}}}(o_{i,j} \mid q, v)} A_{i,j}, \ \text{clip}\left( \frac{\pi_\theta(o_{i,j} \mid q, v)}{\pi_{\theta_{\text{old}}}(o_{i,j} \mid q, v)}, \ 1-\epsilon, \ 1+\epsilon \right) A_{i,j} \right) \right]$$

(3)

with the advantage $A_{i,j}$ defined as:

$$A_{i,j} = \frac{r_{i,j} - \text{mean}(\{r_{i,j}\}_{i=1,j=1}^{G,M})}{\text{std}(\{r_{i,j}\}_{i=1,j=1}^{G,M})}.$$

(4)

We explore three types of transformations $\{\mathcal{T}_i\}_{i=1}^M$ that provide alternative variants of a video: deterministic temporal grouping, stochastic spatial and temporal grouping, and semantic grouping based on per-frame text-to-visual importance scores, which we term importance-based grouping. For deterministic temporal grouping, each $\mathcal{T}_i$ applies the same strided sampling procedure but with different starting offsets: variant $i$ starts at offset $i$ and selects every $M$-th frame, creating complementary temporal views that collectively cover the entire video. For stochastic grouping, each $\mathcal{T}_i$ applies the same augmentation procedures but with independent random sampling. The importance-based grouping will be described in the following paragraph. Each transformation stochastically combines temporal and spatial augmentations (applied with probabilities $p_{\text{temporal}}$ and $p_{\text{spatial}}$), where temporal augmentation performs random temporal cropping selecting 60-90% of frames followed by random stride sampling (stride 1-3), and spatial augmentations include random color jitter and affine transformations.

**Importance-Based Grouping** Our semantic transformation leverages question information to construct semantically-aware groups. Specifically, given a question $q$ and a video $v$ with $F$ frames, we compute a per-frame importance score that measures the relevance of each frame to the question. To this end, we employ a small multimodal model, in our experiments LLaVA-OneVision 0.5B, to extract cross-modal embeddings. Each frame $v_f$ is encoded by the model's vision encoder and mapped into the multimodal embedding space:

$$z_f = \text{Pool}(\text{Proj}(\text{VisionEncoder}(v_f))), \quad z_f \in \mathbb{R}^d,$$

where the Pool operation performs average pooling over the spatial dimensions to obtain a compact frame-level representation. The question $q$ is embedded by averaging its token embeddings:

$$z_q = \frac{1}{L} \sum_{l=1}^{L} \text{Embed}(q_l), \quad z_q \in \mathbb{R}^d.$$

We then normalize both representations, $\hat{z}_f = z_f / \|z_f\|$ and $\hat{z}_q = z_q / \|z_q\|$, and define the importance score for frame $f$ as the cosine similarity:

$$s_f = \hat{z}_f^\top \hat{z}_q, \quad f = 1, \ldots, F.$$

The resulting scores $\{s_f\}_{f=1}^F$ capture fine-grained semantic relevance between the question and video frames, enabling the formation of groups that emphasize the most informative temporal segments. Using these importance scores, we apply segment-wise sampling for group formation. We partition the video into $K$ temporal segments of equal size, where $K$ is randomly sampled between 2 and 6, and select one representative frame from each segment. For a segment with frames $\{v_t\}_{t=s_i}^{e_i}$ and scores $\{s_t\}_{t=s_i}^{e_i}$, we sample the representative frame index $t_i$ using a weighted sampling according to:

$$P(t) = \frac{\exp(s_t)}{\sum_{u=s_i}^{e_i} \exp(s_u)}, \quad t_i \sim \text{Categorical}(P).$$

(5)

This produces an ordered set of indices $\{t_i\}_{i=1}^K$ that ensures temporal coverage while prioritizing question-relevant frames. By introducing question-guided temporal variation across groups, the resulting reward distribution becomes more diverse, which in turn provides the reinforcement learning algorithm with more informative advantage estimates.

| Model | VideoMME (wo sub) | VSI-Bench | PerceptionTest (mc) | TempCompass | VideoMMMU | MMVU (mc) | avg |
|---|---|---|---|---|---|---|---|
| GPT-4o (Achiam et al., 2023) | 71.9 | 34.0 | - | 75.15 | 61.2 | 75.4 | - |
| Gemini-1.5-Pro(Gemini Team, 2024) | 75.0 | 48.8 | - | 70.5 | - | - | - |
| VILA-1.5-8B (Lin et al., 2024) | - | 28.9 | - | - | 20.8 | - | - |
| LongVA-7B (Zhang et al., 2024) | 52.6 | 29.2 | - | - | 23.9 | - | - |
| LLaVA-OV-7B (Li et al., 2024) | 58.2 | 32.4 | 57.1 | 69.5 | - | 64.7 | - |
| Qwen2.5-VL-7B-SFT (Feng et al., 2025) | 55.4 | 33.3 | 68.4 | 69.9 | 49.4 | 63.5 | 56.6 |
| Arrow RL* (Xue et al., 2025) | 60.9 | 36.1 | 66.8 | 72.5 | 47.9 | 62.4 | 57.7 |
| Video-R1 (Feng et al., 2025) | 59.3 | 35.8 | 67.8 | 73.2 | 52.3 | 63.8 | 58.6 |
| Qwen2.5-VL-7B | 56.6 | 34.3 | 68.2 | 72.8 | 48.6 | 57.6 | 56.3 |
| + GRPO | 61.1 | 35.9 | 67.7 | 75.1 | 49.5 | 64.6 | 58.4 |
| + ST-GRPO | 61.7 | 36.6 | 69.4 | 75.1 | 50.1 | 65.4 | **59.7** |

Table 1: Performance on six video understanding benchmarks. ST-GRPO consistently outperforms GRPO, yielding the highest average score with 1000 RL training steps. *Results marked with * are from publicly available model weights.

## 5 EXPERIMENTS

**Benchmarks** We evaluate our approach across a diverse set of benchmark datasets to assess its effectiveness. Specifically, we conduct experiments on VideoMME (Fu et al., 2024) and Temp-Compass (Liu et al., 2024), which focus on fine-grained temporal reasoning and multimodal event understanding; VideoMMMU (Hu et al., 2025) and MMVU (Zhao et al., 2025), which measure performance on complex video-based question answering with diverse domains and question types; VSI-Bench Yang et al. (2024), which emphasizes video-specific to evaluate visual-spatial intelligence; and PerceptionTest (Patraucean et al., 2023), which serves as a comprehensive evaluation of multimodal perception and reasoning. Together, these benchmarks span temporal, spatial, and semantic dimensions of video understanding, offering a rigorous testbed for evaluating the reasoning capabilities of our method in real-world video question answering tasks. To ensure reliable comparisons, we specifically select benchmarks and sub-tasks with deterministic evaluation protocols, thereby avoiding reliance on proprietary LLM-based evaluators.

**Analysis of Reward Variation** As described in Sec. 3, both GRPO and DR.GRPO suffer from bias in the computation of the advantage. GRPO computes advantages by normalizing rewards within a group, encouraging the model to favor relatively better responses. However, it becomes unstable when reward variance is low, while DR.GRPO stabilizes training but down-weights rare correct responses. Increasing reward variance addresses both issues: it prevents exaggerated updates in GRPO and amplifies valuable signals in DR.GRPO. In video question answering, this can be achieved by forming visual groups from different video variants, which complement text-based groups and enhance learning stability. Fig. 3 shows the standard deviation of the rewards over RL training steps and compares GRPO, DR.GRPO and our proposed method ST-GRPO. DR.GRPO shows higher variation in rewards compared to GRPO, but both suffer from a steady decline over training, whereas ST-GRPO maintains consistently higher variance. This improvement arises because low-variance groups lead to poor credit assignment—when all responses look similar, the gradient signal becomes weak and uninformative. Video QA tasks, however, often hinge on subtle temporal or spatial cues, such as an object's motion across frames or the appearance of a critical detail at a specific time. By constructing spatiotemporal groups, ST-GRPO introduces controlled variations that highlight these cues, enabling the model to perceive different facets of the same video. This not only mitigates the low-variance bias but also encourages more informative exploration during reinforcement learning: the model benefits from both the strongest responses within each group and the diversity of responses across visual perspectives. In this way, ST-GRPO leverages the inherent structure of video data to deliver more stable and effective policy updates for video question answering.

**Main Results** Table 1 presents the main results of our method compared to LMMs baselines and reinforcement learning approaches. We report performance across six benchmarks that collectively evaluate temporal reasoning, spatial understanding, and multimodal perception: VideoMME, VSI-Bench, PerceptionTest, TempCompass, VideoMMMU, and MMVU. We first consider supervised fine-tuning (Qwen2.5-VL-7B-SFT) and reinforcement learning baselines (Arrow RL, Video-R1). Among these, Video-R1 achieves the strongest overall performance with an average score of 58.6,

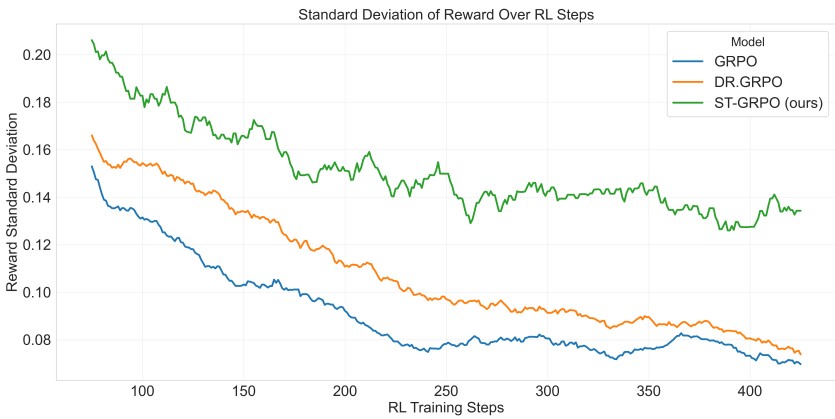

Figure 3: Comparison of the reward standard deviation across GRPO, DR.GRPO, and ST-GRPO over RL training steps. While GRPO and DR.GRPO show a decline in standard deviation over training, ST-GRPO maintains a consistently higher level, indicating greater variability in rewards.

| Method | VideoMME (wo sub) | VSI-Bench | PerceptionTest | TempCompass | VideoMMMU | MMVU (mc) | Avg |
|---|---|---|---|---|---|---|---|
| GRPO | 58.0 | 32.5 | 67.7 | 72.1 | 47.0 | 65.3 | 57.1 |
| ST-GRPO (Deterministic Temporal) | 57.1 | 31.8 | 67.8 | 73.1 | 47.3 | 65.0 | 57 |
| ST-GRPO (Stochastic) | 58.4 | 34.3 | 68.7 | 73.2 | 47.0 | 65.8 | 57.9 |
| ST-GRPO (Importance-Based) | 59.0 | 34.2 | 68.8 | 72.8 | 47.0 | 66.2 | 58.0 |
| ST-GRPO (Importance-Based + Spatial) | 58.2 | 35.2 | 69.3 | 73.1 | 49.3 | 65.4 | **58.4** |

Table 2: Ablation study of ST-GRPO transformation strategies across video understanding benchmarks. ST-GRPO (Deterministic Temporal) employs strided sampling with different offsets, ST-GRPO (Stochastic) applies random temporal cropping and spatial augmentations, ST-GRPO (Importance-Based) uses semantic grouping based on question-frame relevance scores, and ST-GRPO (Importance-Based + Spatial) combines semantic grouping with spatial augmentations.

demonstrating the benefit of reinforcement learning over pure supervised training. Next, we evaluate our reinforcement learning models built on Qwen2.5-VL-7B. Standard GRPO improves upon the base model, raising the average from 56.3 to 58.4, confirming the effectiveness of GRPO in aligning model outputs with video question answering tasks. Our proposed ST-GRPO further boosts performance, achieving a new best average score of 59.7. Notably, ST-GRPO delivers consistent gains across nearly all benchmarks, including VideoMME, VSI-Bench, PerceptionTest, and MMVU. These improvements highlight the benefit of constructing spatiotemporal groups, which increase reward variance and encourage more informative exploration during reinforcement learning.

**Ablation Results** In this section we report the results of the ablation study. For these experiments we conduct 500 steps of RL training.

**ST-GRPO Transformation Variants** We systematically evaluate the effectiveness of different transformation strategies within our ST-GRPO framework through comprehensive ablation experiments. Table 2 presents performance comparisons across six video understanding benchmarks, revealing distinct characteristics of each transformation approach. The deterministic temporal grouping strategy shows mixed results, with marginal improvements on TempCompass (+1.0) but slight decreases on other benchmarks, resulting in comparable average performance (57.0 vs 57.1 for baseline GRPO). In contrast, stochastic transformations demonstrate more consistent improvements, achieving notable gains on VSI-Bench (+1.8) and maintaining competitive performance across all benchmarks with an average of 57.9. The importance-based semantic grouping approach yields strong results, particularly excelling on VideoMME (+1.0) and VSI-Bench (+1.7), achieving an average performance of 58.0. Most significantly, the combination of importance-based grouping with spatial augmentations produces the highest overall performance at 58.4, with substantial improvements on VSI-Bench (+2.7) and VideoMMMU (+2.3). These results demonstrate that semantic-aware transformations, when combined with spatial diversity, provide the most effective enhancement to video understanding capabilities, validating our hypothesis that question-guided frame se-

| Method | VideoMME | VSI-Bench | PerceptionTest | TempCompass | VideoMMMU | MMVU (mc) | Avg |
|---|---|---|---|---|---|---|---|
| GRPO | 58.0 | 32.5 | 67.7 | 72.1 | 47.0 | 65.3 | 57.1 |
| ST-GRPO | 58.2 | 35.2 | 69.3 | 73.1 | 49.3 | 65.4 | **58.4** |
| Dr.GRPO | 59.0 | 32.2 | 68.2 | 72.3 | 47.3 | 64.3 | 57.2 |
| ST-Dr.GRPO | 57.3 | 33.4 | 68.8 | 72.8 | 49.1 | 66.4 | **57.9** |

Table 3: Comparison of GRPO and Dr.GRPO with their spatiotemporal extensions (ST-GRPO and ST-Dr.GRPO). Incorporating spatiotemporal grouping consistently improves average performance across video QA benchmarks.

| M | G | VideoMME | VSI-Bench | PerceptionTest | TempCompass | VideoMMMU | MMVU (mc) | Avg |
|---|---|---|---|---|---|---|---|---|
| 1 | 8 | 58.0 | 32.5 | 67.7 | 72.1 | 47.0 | 65.3 | 55.0 |
| 2 | 4 | 58.2 | 35.2 | 69.3 | 73.1 | 49.3 | 65.4 | **56.2** |
| 4 | 2 | 58.1 | 34.5 | 68.8 | 73.0 | 48.9 | 65.3 | 55.9 |

Table 4: Ablation on the number of generation (G) and (M) number of visual groups.

| Method | VideoMME | VSI-Bench | PerceptionTest | TempCompass | VideoMMMU | MMVU (mc) | Avg |
|---|---|---|---|---|---|---|---|
| GRPO Baseline | 58.0 | 32.5 | 67.7 | 72.1 | 47.0 | 65.3 | 55.0 |
| ST-GRPO (Top-Score Selection) | 57.9 | 35.4 | 68.1 | 73.2 | 47.8 | 64.0 | 55.7 |
| ST-GRPO (Importance-Weighted) | 58.2 | 35.2 | 69.3 | 73.1 | 49.3 | 65.4 | **56.2** |

Table 5: Comparison of sampling strategies within importance-based grouping. ST-GRPO (Top-Score Selection) selects the highest-scoring frame from each segment, while ST-GRPO (Importance-Weighted) uses the probabilistic sampling method described in Eq. 5.

lection complemented by visual robustness leads to superior performance across diverse benchmarks.

**Effect of Spatiotemporal Grouping on GRPO and Dr.GRPO** Table 3 compares the performance of standard GRPO and Dr.GRPO with their spatiotemporal variants (ST-GRPO and ST-Dr.GRPO). Across all evaluated benchmarks, incorporating spatiotemporal grouping yields consistent improvements in average performance. For GRPO, introducing spatiotemporal groups increases the overall accuracy from 57.1 to 58.4, while for Dr.GRPO the average rises from 57.2 to 57.9.

**Comparison of Sampling Methods in Importance-Based Grouping** Table 5 compares different frame sampling strategies within ST-GRPO. While both variants outperform the GRPO baseline, the *importance-weighted* strategy consistently yields higher performance across most benchmarks, achieving the best average score. This shows that probabilistic sampling better balances diversity and semantic alignment than simply selecting top-scoring frames.

**Impact of Number of Generation and Visual Groups** We conduct an ablation study to determine the optimal configuration of generation count ($G$) and visual group count ($M$) in our ST-GRPO framework, while maintaining a constant total computational budget of $G \times M = 8$. Table 4 presents the results across different configurations. The baseline configuration with $M = 1$ (single visual group) and $G = 8$ generations achieves an average performance of 55.0. Increasing the number of visual groups to $M = 2$ with corresponding $G = 4$ generations yields the best performance at 56.2 average, with notable improvements on VSI-Bench (+2.7), PerceptionTest (+1.6), and VideoM-MMU (+2.3). However, further increasing visual groups to $M = 4$ with $G = 2$ generations results in a slight performance drop to 55.9, suggesting diminishing returns from excessive visual group partitioning. These results indicate that a balanced approach with moderate visual group diversity ($M = 2$) provides the optimal trade-off between exploration breadth and generation quality within the same computational constraints.

**Qualitative results** These qualitative results highlight the superior spatiotemporal reasoning of ST-GRPO compared to the baseline GRPO model. In the first example, ST-GRPO correctly identifies the woman's mode of transport as driving a car, grounding its prediction in visual evidence such as the car's interior and the highway scene. In contrast, GRPO incorrectly infers she is walking, misled by frames where she appears on foot at other times in the video, even though these do not indicate

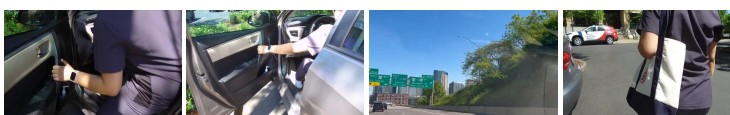

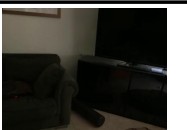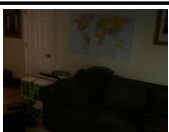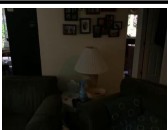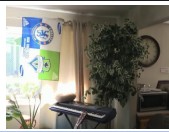

**Q:** What is the lady's mode of transport to work in the video?
A. By driving a car.  B. By bus. C. By subway.  D. By walking.

**GRPO:** The video shows a lady walking towards a building, which suggests she is on foot. There's no indication of her using a car, bus, or subway. The setting looks like a typical office environment, and the presence of a desk and computer implies she's starting her workday. Given these observations, it's clear that the lady is walking to work. <answer>D</answer>

**ST-GRPO:** The video shows a lady driving a car to work. There's a clear view of the car's interior, including the steering wheel and dashboard. The lady is seen driving on a highway, which is a common mode of transport for many people. There's no indication of her using a bus, subway, or walking. The video starts with her driving, so it's reasonable to conclude that this is her mode of transport to work. <answer>A</answer>

**Q:** What will be the first-time appearance order of the following categories in the video: door, pillow, clock, lamp?
A. pillow, door, lamp, clock, **B**. lamp, clock, door, pillow, **C**. door, pillow, clock, lamp

**GRPO:** Let's break it down step by step: 1. Door: The door is typically a fixed object in a room and is usually visible early in the video as the camera pans around the room. 2. Pillow: Pillows are often placed on furniture like sofas or chairs, and they appear later in the video as the camera focuses on specific areas. 3. Clock: Clocks are usually small and can be placed on tables or shelves, appearing later in the video. 4. Lamp: Lamps are also small and can be placed on tables or shelves, appearing later in the video. <answer>C</answer>

**ST-GRPO:** The video starts with a view of a room with a couch and a window. The first object I see is a pillow on the couch. Then, the camera moves to show a door, which is the next object. After the door, the camera focuses on a lamp, which is on a table. Finally, the camera moves to show a clock on the wall. So, the order of appearance is: pillow, door, lamp, clock. <answer>A</answer>

Figure 4: **Qualitative results** A comparison of our ST-GRPO model with the baseline GRPO model. (Top) Our approach correctly answers a question about a person's mode of transport. (Bottom) It also correctly determines the temporal order of objects, demonstrating a stronger ability to reason about spatiotemporal relationships.

she is commuting to work by walking. In the second example, ST-GRPO accurately determines the first-time appearance order of objects by following the visual sequence in the video, whereas GRPO fails by relying on a flawed object-based prior rather than the actual temporal progression. These cases illustrate ST-GRPO's ability to anchor its reasoning in relevant visual and temporal signals. We hypothesize that its importance-based grouping mechanism—leveraging question-specific information to guide frame sampling—enables a more coherent understanding of temporal progression by focusing on the most relevant objects, persons, and actions mentioned in the query.

## 6 CONCLUSION AND FUTURE WORK

We presented **SpatioTemporal-GRPO (ST-GRPO)**, a reinforcement learning framework that extends Group Relative Policy Optimization by incorporating spatiotemporal diversity into video question answering. By generating multiple variants of input videos and introducing an importance-based strategy to prioritize frames most relevant to the query, ST-GRPO increases reward variance and provides more informative learning signals. Experiments across six challenging benchmarks demonstrate consistent improvements over GRPO and Dr.GRPO, underscoring the value of explicitly leveraging video structure in post-training. Future directions include scaling to longer videos with more advanced frame selection, exploring adaptive weighting schemes for spatiotemporal variants, and extending the approach to other video-language tasks such as dense captioning, retrieval, and video-grounded reasoning, moving toward spatiotemporal reinforcement learning as a general paradigm for multimodal training.

## 7 REPRODUCIBILITY STATEMENT

We will make our code available upon acceptance. For all experiments, publicly available models and datasets have been used. We did not use any data for training that is not publicly available.

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

# A    APPENDIX

**Implementation**    We use Qwen2.5-VL-7B as the base LMM for reinforcement learning exper-
iments. For computational efficiency, we limit inputs to a maximum of 16 video frames during
training and process frames at a maximum resolution of $128 \times 28 \times 28$ pixels and and response
length is capped at 768 tokens. The model is trained for only 1000 RL steps on 4 GPUs using the
video data of Video-R1-data (Feng et al., 2025). For our experiments, we set the hyperparameter $\beta$
in the KL divergence term to 0.04. We used the Adam optimizer with a learning rate of 1e-6. For a
fair comparison with the baseline model, which uses $G = 8$ generations in its Group Relative Policy
Optimization (GRPO) algorithm, we configured our ST-GRPO model with $G = 4$ and $M = 2$. This
setup results in a total of $G \times M = 8$ groups, ensuring an equivalent number of groups for both
models. Following (Feng et al., 2025; Guo et al., 2025), we use accuracy and format rewards in our
RL training.

