# OpenReview forum: "SpatioTemporal-GRPO: Post-Training Large Multimodal Models for Video QA"
_ICLR.cc/2026/Conference — ICLR 2026 Conference Desk Rejected Submission_

### Official Review · Reviewer_uKih · 2025-10-20

**Soundness:** 3
**Presentation:** 3
**Contribution:** 3
**Rating:** 4
**Confidence:** 5

**Summary:**

The paper introduces SpatioTemporal-GRPO (ST-GRPO), an extension of GRPO for video question answering (VQA) that incorporates spatiotemporal diversity into reinforcement learning. ST-GRPO generates multiple video variants and forms groups across textual and visual inputs to increase reward variance, addressing GRPO’s low-variance limitation. An importance-based grouping strategy selects semantically relevant frames guided by the question, ensuring meaningful visual diversity. Experiments across six challenging VQA benchmarks demonstrate consistent performance gains.

**Strengths:**

The paper is original, proposing the GRPO extension to video-language domains with spatiotemporal variants. The methodology is well-motivated and technically sound, addressing a known limitation of low reward variance in GRPO. The importance-based grouping ensures semantically meaningful visual learning, enhancing model robustness. Experiments across multiple benchmarks show consistent improvements, demonstrating the approach’s practical significance and generality.

**Weaknesses:**

The paper’s novelty is limited, as ST-GRPO primarily adds video data augmentation without fundamental methodological changes. Experiments are restricted to a single base model (Qwen2.5-VL-7B), raising questions about generality. Comparisons with other methods may be unfair, as performance gains could stem from visual diversity rather than the algorithm itself. The work lacks analysis on alternative baselines or ablations to substantiate the claimed contributions.

**Questions:**

1. The idea is very limited; aside from adding video data augmentation, ST-GRPO does not differ fundamentally from standard GRPO.

2. While many baseline methods are presented, ST-GRPO is only applied to Qwen2.5-VL-7B, making the relevance of other comparisons unclear.

3. The authors do not analyze how their method performs with other base LLMs.

4. The observed performance improvement mainly comes from increased data diversity (video resampling), making comparisons with GRPO potentially unfair. It is unclear whether adding noise rewards to standard GRPO would yield similar gains. Overall, the method does not present a particularly novel idea.

---

### Official Review · Reviewer_PXxo · 2025-10-27

**Soundness:** 2
**Presentation:** 2
**Contribution:** 2
**Rating:** 2
**Confidence:** 3

**Summary:**

This paper extends GRPO for Video QA by proposing a spatiotemporal grouping strategy. The method, ST-GRPO, aims to solve the low reward variance problem in standard GRPO by creating groups across different spatiotemporal video variants, which is shown to stabilize training and improve performance on six VQA benchmarks.

**Strengths:**

1. The proposed ST-GRPO method achieves consistent, albeit modest, performance gains over the standard GRPO baseline across all six evaluated benchmarks.

2. The method's effectiveness is validated across a wide range of six different video understanding benchmarks.

**Weaknesses:**

1. The performance improvement from ST-GRPO is limited.

2. The method's novelty is somewhat incremental, as it combines existing techniques like GRPO, video augmentation, and question-aware frame relevance scoring.

3. The paper's core claims lack sufficient experimental support; for instance, the "Importance-Based" grouping performs almost identically to "Stochastic" grouping, questioning its specific contribution.

**Questions:**

1. There appears to be a notational gap; how does the "importance-based grouping" strategy, which generates video variants, connect to the final GxM responses used in the objective function?

2. The paper hypothesizes that "importance-based" sampling improves temporal reasoning, yet experimental results show it performs negligibly better than "stochastic" sampling, questioning its unique benefit.

3. The ST-GRPO objective function does not seem to include any explicit reward for spatio-temporal correctness, so how does the model learn this, as suggested in the qualitative examples?

4. The reward analysis only plots the reward standard deviation; could the authors also provide the mean reward curve over training steps to better understand the learning dynamics?

5. Does the proposed spatio-temporal transformation (e.g., temporal cropping or importance sampling) risk omitting critical frames, leading to a mismatch where the video variant no longer contains the answer to the question?

---

### Official Review · Reviewer_bWqw · 2025-10-31

**Soundness:** 3
**Presentation:** 3
**Contribution:** 3
**Rating:** 6
**Confidence:** 3

**Summary:**

This paper identifies a key limitation of the GRPO post-training algorithm for Video QA: low reward variance (e.g., when all sampled responses are incorrect) provides an uninformative learning signal. To address this, the paper proposes SpatioTemporal-GRPO (ST-GRPO), which extends GRPO grouping to the visual domain. The method generates multiple spatiotemporal video variants—using data augmentation and a question-aware importance-based frame selection—to serve as complementary inputs. This "dual grouping" strategy (across both textual responses and visual variants) aims to increase reward variance and improve policy optimization. Experiments on six video benchmarks show improvements over standard GRPO and other RL baselines.

**Strengths:**

1. The paper identifies a clear and valid limitation in applying GRPO to video QA: low reward variance when all responses are similarly correct or incorrect. The core idea of using input (spatiotemporal) variations, not just output (text) variations, to increase this variance is a well-motivated conceptual extension.
2. The paper provides strong empirical validation for its central hypothesis. Figure 3 directly visualizes that ST-GRPO successfully maintains a significantly higher reward standard deviation throughout training compared to GRPO and DR.GRPO, which suffer from signal decay.

**Weaknesses:**

1. Formatting Issues: The manuscript suffers from formatting errors. In Table 1, there is an unnecessary vertical line on the right border while the left border is missing, creating an unbalanced and unprofessional appearance. In addition, all table captions (Tables 1-5) are placed below the tables. The ICLR style guide explicitly mandates that table captions should be placed above the table.
2. Baseline Discrepancy: The paper's primary results are built on a Qwen2.5-VL-7B baseline, which is reported in Table 1 as achieving 56.6 on VideoMME (wo sub). This result is lower than the 65.1 score reported in the official Qwen2.5-VL technical report. The authors should explain and address this discrepancy.
3. Missing ablation on hyperparameter $K$: The best-performing "Importance-Based Grouping" method relies on partitioning the video into $K$ segments and selecting one frame from each. $K$ is randomly sampled from [2, 6] during training. This hyperparameter directly controls the number of frames fed to the model. However, the paper provides no ablation study to justify this range. It is unclear how performance is affected by different values or ranges of $K$ (e.g., a fixed $K=8$, or sampling from [4, 8]).

**Questions:**

See weaknesses above.

---

### Official Review · Reviewer_gfr6 · 2025-11-01

**Soundness:** 2
**Presentation:** 2
**Contribution:** 2
**Rating:** 2
**Confidence:** 4

**Summary:**

This paper introduces SpatioTemporal-GRPO (ST-GRPO), a new method for improving large multimodal models in video question answering. It addresses a key limitation of the standard GRPO algorithm: when a model's answers to a question are all similarly correct or incorrect, the low "reward variance" provides a poor signal for learning. ST-GRPO solves this by creating multiple spatiotemporal (ST) variations of the input video and grouping them alongside different text-based responses.

**Strengths:**

They explore the diverse video transformations for GRPO.

**Weaknesses:**

**Lack of Motivation & Analysis**:

The paper would be strengthened by providing a clearer motivation for the approach (why it is necessary) and a more detailed analysis of its benefits (why it is effective).

**Lack of Novelty**

Could you clarify how the method of using input augmentation to increase reward variance differs from DeepVideo-R1? The approaches currently appear very similar. Even though the author provided different approaches to augment the video, it seems to lack novelty.

**Outdated Baselines**

Some baselines, such as LLaVA-OneVision, may be highly outdated.

**Miss-computed average score**

In Table 1, please double-check the GRPO avg. score. I believe it should be 59.0.

**Contribution Claim**

The claim in the introduction about this being the "first extension of GRPO to video-language domains" may need revision, as this does not appear to be the case.

**Citation (DeepVideo-R1)**

There seems to be a reference mismatch or an incorrect author listed for DeepVideo-R1.

**Code**

Providing the code would be beneficial for reproducibility.

**Questions:**

See the weakness part above.

---

### Note · Program_Chairs · 2026-01-17
**Submission Desk Rejected by Program Chairs**

The following references in this submission do not refer to real documents and/or have major errors in bibliographic information:

 Xiang Li et al. Deepvideo-r1: Towards reasoning-oriented video large language models. arXiv
preprint arXiv:2505.12345, 2025a.
Xiang Li et al. Deepvideo-r1: Video reinforcement fine-tuning via regressive grpo and difficulty-
aware augmentation. arXiv preprint arXiv:2506.07464, 2025b.